# The Cow’s Milk-Related Symptom Score (CoMiSS^TM^): Health Care Professional and Parent and Day-to-Day Variability

**DOI:** 10.3390/nu12020438

**Published:** 2020-02-09

**Authors:** Yvan Vandenplas, Eva Carvajal, Stefaan Peeters, Nadine Balduck, Yesra Jaddioui, Carmen Ribes-Koninckx, Koen Huysentruyt

**Affiliations:** 1KidZ Health Castle, UZ Brussel, Brussels, Vrije Universiteit Brussel, 1090 Brussels, Belgium; Nadine.balduck@skynet.be (N.B.); koen.huysentruyt@uzbrussel.be (K.H.); 2Department of Paediatrics, Hospital Casa de Salud, 46021 Valencia, Spain; eva.carvajal2@gmail.com; 3Aalsters Stedelijk Ziekenhuis, 9300 Aalst, Belgium; stefaanpeeters@me.com; 4Department of Pediatric Gastroenterology, Hepatology and Nutrition, La Fe University Hospital, 46026 Valencia, Spain; ribes_car@gva.es; 5Division of Pediatric Gastroenterology, Hepatology and Nutrition, Department of Pediatrics, The Hospital for Sick Children, University of Toronto, Toronto, ON M5G 1X8, Canada

**Keywords:** Cow’s Milk-related Symptom Score, CoMiSS, infant

## Abstract

The Cow’s Milk-related Symptom Score (CoMiSS^TM^) was created as an awareness tool for cow’s milk allergy. The aim of the present study was to analyze the inter-rater variability between a pediatrician, parents, and day to day variability. A Health Care Professional (HCP) and parent filled in the CoMiSS independently and blinded for each other to evaluate inter-rater variability. In order to validate day-to-day variability, a parent filled in the CoMiSS during 3 consecutive days and was compared to the CoMiSS scored by the HCP. The absolute agreement between parent and HCP was 75%, and 92.6% and 100% with a tolerance of 0, 1, and 2 points, respectively, resulting in excellent agreement with an intraclass correlation coefficient (ICC) 0.981 (95% Confidence Interval 0.974–0.986, *p* < 0.001). Day-to-day variability during 3 consecutive days resulted in an absolute agreement of 30%, increasing to 80% and 88.6% when 2 and 3 points, respectively, were accepted. The ICC was excellent for the parental prospective scores (0.93, 95% CI 0.90–0.96; *p* < 0.001). Day-to-day variability indicates that CoMiSS has a moderate inter-rater reliability. A very low variability was observed when scored prospectively over three days. Data suggest that the CoMiSS can reliably be scored by parents without additional training.

## 1. Introduction

The Cow’s Milk-related Symptom Score (CoMiSS^TM^) was created as an awareness tool for cow’s milk protein allergy (CMPA) in infants [1]. It is a symptom based score with the additional aim of evaluating evolution during a therapeutic intervention. As most of the symptoms in the CoMiSS are nonspecific on themselves, we recently investigated the performance of the CoMiSS in a large cohort of presumed healthy infants [2]. This resulted in a median value of 3 for healthy infants and a 95th centile of 9, which led us to propose a change in cut-off value from ≥12 to >10 for a ‘positive’ CoMiSS [2]. Polish data are in alignment with these findings as the median (Q1–Q3) and mean (SD) CoMiSS™ values were 4 (2–7) and 4.7 (3.5), respectively [3]. The 95th percentile was 11. Eleven children out of 226 (4.9%) scored >12 [3]. However, there are no data (i) on the inter-rater variability between a health care professional (HCP) and parents and (ii) on day-to-day variability. Therefore, we investigated the inter-rater reliability of the CoMiSS between parents and health care professionals (HCPs), and the day to day variability of the CoMiSS rated by a HCP and parents on four consecutive days.

## 2. Methods

Ethical approval was obtained from the Ethical Committee of the UZ Brussel (Protocol 16. NHS; B.U.N. 143201730871). The original CoMiSS was used, which has a range from 0 to 33 (Table 1).

### 2.1. HCP versus Parents

This part of the study was performed in Spain between October 2017 and February 2018. Parents attending a primary HCP for a routine scheduled visit or a pre-vaccination assessment with a presumed healthy infant (2–6 months old) were informed about the CoMiSS. Inclusion criteria were: no complaints uttered by the parents during the interview with the pediatrician; exclusion criteria were: any health problem detected by the pediatrician during the interview or physical examination. Pediatricians scored the CoMiSS based on the history provided by the parents and on the physical exam. After signing the informed consent to participate, the HCP filled in the CoMiSS. After the routine visit, the parent—who was blinded to the pediatrician’s initial total score—was asked to fill in a second CoMiSS in the waiting room.

### 2.2. Four Days CoMiSS Variability

This part of the study was performed in Belgium between December 2017 and June 2018. A CoMiSS was obtained on the same infant (<6 months old) by a general pediatrician according to the information obtained from a parent during a routine follow-up consultation based on preceding days (t_1_). The parent was blinded to the pediatrician’s CoMiSS. The CoMiSS was then prospectively recorded by the same parent during the following three days (t_2_–t_4_). The same inclusion and exclusion criteria as for the Spanish cohort were applied. Parents were instructed to not change the dietary intake during these four days.

### 2.3. Statistical Analysis

For the Spanish cohort, CoMiSS obtained by the pediatrician and the parent were compared. Comparisons were made between the HCP (t_1_) and parent (on t_2_) CoMiSS and between the parental CoMiSS recorded on three consecutive days (t_2_–t_4_) for the Belgian cohort. Differences in distributions were compared using a paired Wilcoxon signed-rank test, as the data was not normally distributed. There were two missing observations for the Belgian cohort on t_4_.

All scores were compared as a total score, and as the number of children who scored positive on the original scale (≥12) and on the revised scale (>10). Percentage of absolute agreement with 0–3 points tolerance and estimates for the intra-class correlation coefficient (ICC) with their 95% confidence intervals were calculated for all the comparisons mentioned earlier. The ICC estimates were based on a “two-way” random model, using single agreements for HCP versus parent comparisons and average agreements for parental day to day variability; a *p*-value < 0.05 was considered significant. Analyses were performed using R version 3.4.3, with the “irr” package for calculation of the ICC estimates and their confidence intervals (CI). Based on the value of the ICC, agreement was categorized as poor (ICC < 0.50), moderate (0.50 ≤ ICC < 0.75), good (0.75 ≤ ICC < 0.90) or excellent (ICC ≥ 0.90) [4]. A “moderate” ICC value was considered to be indicative of an acceptable level of agreement. We hypothesized that there would be an acceptable level of reliability for the CoMiSS, both when considering day to day variability and when comparing HCP with parental scores.

## 3. Results

### 3.1. HCP versus Parent

One hundred forty-eight Spanish infants were included, with a median (Inter-Quartile Range) age of 2.3 (2.9) months. No statistically significant difference was observed for any item of the CoMiSS or the total score, depending on the person performing the scoring, parent versus pediatrician (Table 2). Feeding, breastfeeding o formula feeding, had no influence on the CoMiSS. When scored by parents, eight (5.4%) infants had a score >12, which was initially proposed as the cut-off (1) and 12 (8.1%) infants had a score >10, which came up as the best cut-off according to the data obtained in presumed healthy infants (2). Similar results were seen when scored by HCP: seven (4.7%, *p* = 0.791) scored ≥12 and 11 (7.4%, *p* = 0.828) scored >10. This indicates that only one child had a change in CoMiSS outcome when rated by a parent compared to a HCP. The high scores (>10) given by the pediatrician or parent are listed in Table 3. If a score of >12 is used as cut-off, only one child changes from positive to negative score, as the score given by the parent was 12 and by the pediatrician 11. If a cut-off of >10 was applied, no infant changed from their category.

The absolute agreement was 75%, 92.6%, and 100% with a tolerance of 0, 1, and 2 points, respectively, resulting in an excellent agreement based on the ICC estimate: 0.981 (95% CI 0.974–0.986, *p* < 0.001). Thirteen infants had a CoMiSS ≥10, with 8 of them scoring ≥12; a final diagnosis of CMPA confirmed by a food challenge was established in, respectively, 10 out of the 13 (76%) and in 7 of the 8 infants with a CoMiSS >12.

### 3.2. Four Days CoMiSS Variability

Seventy-two Belgian infants were enrolled in this part of the study (52% male; mean (SD) age 3 (0.5) months). The median (IQR) CoMiSS score at t_1_ (by HCP) was 3.7 (5.0); whilst median (IQR) scores by a parent on t_2_–t_4_ were respectively 3.0 (4.0), 3.0 (4.0) and 2.0 (4.0); all pairwise comparisons were not significantly different (unadjusted *p* values between 0.26 and 0.82). None of the subcategory scores had a significant difference in any of the pairwise comparisons (all unadjusted *p* values ≥ 0.32). At t_1_ only one (1.4%) infant had a positive score by either cut-off at t_1_ (CoMiSS was 12). This particular infant had however consistently scored <10 when rated by the parent). One infant with a negative score at t_1_ was scored positive at t_2–4_ by the parent.

The absolute agreement between HCPs and parents on t_1_ vs. t_2_ was 25%, this increased to 68.1% when a tolerance of 2 points was accepted and 77.8% when a tolerance of 3 points was accepted. The absolute agreement for the CoMiSS by a parent on t_2_–t_4_ (Figure 1) was 30%, which increased to 80% and 88.6% when a tolerance of respectively 2 and 3 points were accepted. The ICC estimate for retrospective vs prospective (t_1_–t_2_) was moderate (0.53; 95% CI 0.34–0.68; *p* < 0.001) and excellent for the prospective scores by a parent over 3 days (0.93 (95% CI 0.90—0.96; *p* < 0.001).

Legend: The dotted line represents the old cut-off values of ≥12 for a positive score, the dashed line the new cut-off of ≥10 for a positive score. This figure represents the individual infants in the X-axis, the Y-axis represents the total CoMiSS as scored by the parents on t_2_–t_4_. The longer the vertical line per patient is, the higher the variability is for that individual patient over 3 days. It shows that the variability happens mainly in the scores between 3 and 9, and that only one infant had a score that changed from 9 to 10.

## 4. Discussion

The agreement between a CoMiSS in presumed healthy infants provided by a pediatrician or parents was excellent, without the need to provide any special training to the parents. This means that the CoMiSS can be recorded in a reliable way by parents at home or in the waiting room, preceding a consultation. It is possible that by performing the scoring at home, a larger difference may be observed due to a longer time span between recording by the parents and the consultation with the pediatrician. According to the analysis of the data obtained in a population of presumed healthy infants, a score of >10 was proposed as the best cut-off value [2]. A recent Italian study evaluated the CoMiSS in symptomatic infants, proposing a cut-off of >9 [5]. According to the data from Poland, the 95th percentile was 11 [3]. The receiver operation characteristic curve identified a CoMiSS of 9 to be the best cut-off value (84% sensitivity, 85% specificity, 80% positive (PPV), and 88% negative predictive value (NPV)) for the response to a cow milk free diet [5]. Whatever cut-off considered, only 1 patient in the Spanish cohort changed from positive to negative i.e., 1 out of 13 for ≥10 limit. Moreover, the majority (76%) of the infants scoring 10 or above had a final diagnosis of CMPA, confirmed by a challenge test.

The day-to-day variability indicates that the CoMiSS score has a moderate inter-rater reliability if used retrospectively vs. prospectively. A very low variability was observed when scored prospectively over three days by the same assessor.

A likely explanation for the poorer performance of Belgian pediatricians vs. parent assessment when compared to the Spanish parents is the difference in how the data was obtained. In the Spanish cohort, both assessors were asked to fill in the CoMiSS retrospectively, while in the Belgian data there were two sources of variability between t_1_ and t_2_: retrospective vs prospective rating and a different rater.

CoMiSS was reported to help in predicting CMPA in children aged less than 2 years in an Indian primary care setting, aiding in early diagnosis [6]. The next step is to confirm these data in symptomatic infants. According to the Spanish data, the indication is that the variability between parent and pediatrician is also very low for high scores.

In conclusion, the CoMiSS has a moderate inter-rater reliability if used retrospectively vs prospectively. A very low variability was observed when scored prospectively over three days by the same assessor or when scored retrospectively by a parent and pediatrician. These data suggest that parents can score the CoMiSS reliably prior to a consultation without special training.

## 5. Conclusions

A very low variability was observed when the CoMiSS is scored prospectively over three days. Parents and health care providers score the CoMiSS in a very similar way. Our data indicate that the CoMiSS can reliably be scored by parents without additional training.

## Figures and Tables

**Figure 1 nutrients-12-00438-f001:**
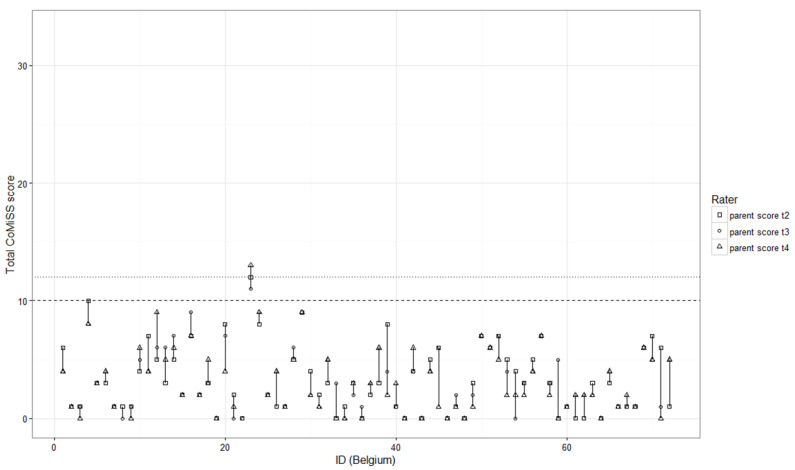
Day to day variability in individual CoMiSS on three consecutive days by a parent.

**Table 1 nutrients-12-00438-t001:** Cow’s Milk related Symptom Score (CoMiSS^TM^).

Symptom	Score			
Crying (°)	0	<1 h/day		
	1	1–1.5 h/day		
	2	1.5–2 h/day		
	3	2–3 h/day		
	4	3–4 h/day		
	5	4–5 h/day		
	6	>5 h/day		
Regurgitation	0	0–2 episodes/day		
	1	>3–<5 of small volume		
	2	>5 episodes of >1 coffee spoon		
	3	>5 episodes of + half of the feed in < half of the feeds		
	4	continuous regurgitations of small volumes >30 min after each feed		
	5	regurgitation of half to complete volume of a feed in at least half of the feeds		
	6	regurgitation of the complete feed after each feeding		
Stools (Bristol scale)	4	type 1 and 2 (hard stools)		
	0	type 3 and 4 (normal stools)		
	2	type 5 (soft stool)		
	4	type 6 (liquid stool, if unrelated to infection)		
	6	type 7 (watery stools)		
Skin symptoms	0 to 6	Atopic eczema		
			Head/neck/trunk	Arms/hands/legs feet
		Absent	0	0
		Mild	1	1
		Moderate	2	2
		Severe	3	3
	0 or 6	Urticaria (no 0/yes 6)		
Respiratory symptoms	0	no respiratory symptoms		
	1	slight symptoms		
	2	mild symptoms		
	3	severe symptoms		

Legend (§) Although many infants with cow’s milk related symptoms have no impaired growth or weight gain, faltering of these parameters suggests organic disease, of which cow milk protein allergyis a possible cause. (°) Crying was only considered if the child was crying for one week or more, assessed by the parents, without any other obvious cause.

**Table 2 nutrients-12-00438-t002:** Comparison the CoMiSS by a trained health care professional versus parent in a Spanish cohort of presumed healthy infants.

	HCP Median (IQR)	Parent Median (IQR)	*p*-value
Crying	0.0 (2.0)	0.0 (2.0)	0.091
Regurgitation	0.0 (1.0)	0.0 (1.0)	0.080
Stools consistency	2.0 (2.0)	2.0 (2.0)	0.773
Skin symptoms	0.0 (0.0)	0.0 (0.0)	1.000
Respiratory symptoms	0.0 (0.0)	0.0 (0.0)	1.000
Total CoMiSS	4.0 (5.0)	4.0 (4.2)	0.069

**Table 3 nutrients-12-00438-t003:** Individual CoMiSS by parent or pediatrician with a positive score in the Spanish cohort.

Parent	Pediatrician
17	16
14	14
13	12
14	13
13	13
12	12
11	11
11	11
10	9
10	10
10	10

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
