# Peer review of "The Cow’s Milk-Related Symptom Score (CoMiSSTM): Health Care Professional and Parent and Day-to-Day Variability"

_nutrients, 2020, doi:10.3390/nu12020438_

Round 1

Reviewer 1 Report

Clarify in methodology the guidelines for diet for the infants during the timeframes that parents were doing the Comiss scores. It is important to show that both the Belgian and the Spanish cohorts were given similar instructions to maintain usual feeding practices/daily care to address colic, etc. Also, it would be worthwhile to mention the percentage of infants in all cohorts breastfed exclusively/ bottle fed exclusively just to show that the COmiss tool is reliable among raters retrospectively regardless of infant feeding practices. Lastly, clarify if the pediatricians were basing their Comiss score solely on the history obtained in the visit from the parent or adding physical exam findings (eczema, hives, wheezing) as well.

Reviewer 2 Report

Extensive english language edition required. Some of the suggestions are mentioned below: 

Abstract: place "period" without any space after "allergy".  If you are using "paediatrician" in abstract, use same for rest of the text also; at some places, you have used "pediatrician".  Abstract: parent filled in a blinded, CoMiss and not "Comiss, blinded". After 3 consecutive days, insert "and was"compared In the absolute agreement statement, 75%, 92.6% and 100% with a tolerance of 0,1 and 2 points respectively Day to day variability: 80% and 88.6% when 2 and 3 points respectively, were accepted.  There are extra "space" characters in abstract. Omit.  Introduction: You abbreviated CMPA but there is no protein in the expansion Introduction: themselves and not their selves Use CMPA in the legend of Table 1 Statistical analysis: Differences not difference; use children who scored positive, not children scoring positive  In Results, expand IQR Statement below Table 3: Absolute agreement: change as per comment # 4 Section 3.2: Use HCP and not MD as you have been using HCP throughout Discussion: 2nd line: change to "without the need for" instead of without providing; line 4: change to "It is possible that performing the scoring at home, a larger difference may be observed due to a longer time span between" Discussion: After "a cut-off of >=9", the reference 5 is in superlative; instead put in inside square brackets; there is an underscore under a period after limit- remove Discussion: First paragraph: change to "confirmed by a challenge test".  Discussion: place hyphen in cut-off.  Remove extra spaces throughout the text Second last paragraph: Change to help in predicting CMPA Remove reference #3, unpublished article is not helpful. 

Reviewer 3 Report

Dear Editor and Authors

Scoring CMA related symptoms especially in infants it is problematic from always. There is a thin line of trust between the physician and parents. Parents are usually seen as persons who over-interpret the symptoms in their child, especially when they come from an infant or small one and not talking yet. So somebody who put together a few simple symptoms observed by each parent to monitor baby everyday conditions and do statistical analysis to prove its independence from the observer (HCP vs parent) do a great job.
This 148 infants group for this kind of study could be extended probably. Anyway to start looking for something new it is enough.
In addition, I have to say that the manuscript is very nicely written - just good for reading. To truly say I didn't have a manuscript like this to review for a long time.
So finally I approve of publishing in the form as it is.
